# The Paradox of the Payday Borrower: A Case Study of the Role of Planned Behavior in Borrowers' Motivations and Experiences

**Irene Herremans** [1,*], **Peggy Hedges** [1], **Fereshteh Mahmoudian** [2], **Anne Kleffner** [1] 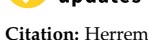 and **Mahrukh Tahir** [1]

1    Haskayne School of Business, University of Calgary, Calgary, AB T2N 1N4, Canada
2    Beedie School of Business, Simon Fraser University, Burnaby, BC V5N 1S6, Canada
*    Correspondence: irene.herremans@haskayne.ucalgary.ca

**Abstract:** This research used the theory of planned behavior as a framework to investigate the role of attitudes, behavioral control, norms, and previous behavior in payday loan borrowers' difficulty or lack of difficulty in repaying loans. The data were collected from 138 respondents with payday loan experience via a questionnaire in a city in a western province in Canada as part of a campaign to change payday loan regulations. The research findings show that different approaches are necessary to address the needs of distinct types of payday borrowers, based on their repayment abilities and whether the loan improved their quality of life in the long term. Furthermore, we found, similar to previous literature, a group of payday borrowers who lack financial confidence. This sub-group is referred to as the "unsure" sub-group in our research and provides opportunities to improve the payday learning context. To accommodate the unsure group, payday lenders and conventional financial institutions can collaborate to offer innovative financial instruments, improve financial literacy through education, and provide better access to information about borrowers' financial status. The confirmation of this unsure group also leads us to recommend further study to determine opportunities for payday borrowers to become better informed about their options, to increase financial confidence.

**Keywords:** payday loans; borrowers' profiles; theory of planned behavior; financial confidence; financial literary; financial fragility

## 1. Introduction

Since payday lending was legalized in both the United States and Canada in the 1990s (Kobzar 2012b; Conference Board of Canada 2016), the exponential growth of the industry has generated increasing attention from regulators, policy makers, consumer advocacy groups, and other stakeholders. As the market has grown, so have concerns about payday borrowers and the potential negative effects that they may experience. This research study was undertaken to understand the payday loan borrowers' characteristics to better serve the market in a large city in the province of Alberta, Canada. It was related to a campaign to change policy and regulations regarding payday loans, primarily to reduce interest rates and fees, and to lengthen the payment period for payday loans (Bell 2016; Bickis 2017). In the process of the change, some members of the financial community investigated better approaches to serve borrowers who need short-term loans and to ensure financial literacy.

The change in regulation targeted the most financially vulnerable users of payday borrowers. However, our study revealed important characteristics about payday borrowers' characteristics that can help to serve the entire market. A common assumption frequently adopted by the media and policy makers is that all payday borrowers are caught in a vicious, downward spiral once they have taken out their first loan. That is, they all encounter difficulty in repaying a payday loan and find it necessary to borrow a larger sum to repay the original loan, plus the required interest and fees, negatively affecting their quality of life in the long term. Definitely, targeting policy toward the most vulnerable

is critical; however, knowledge of the characteristics of all payday borrowers can lead to better service overall.

Building on past research, we investigate whether there are distinct profiles of the payday loan market. Furthermore, if there are distinct profiles, how are meaningful, effective public policy and regulations developed for each profile, along with well-crafted and targeted educational tools? See Bolen et al. (2020) for a reasonably current summary of the debate surrounding small dollar credit products, including payday loans. The extant research on payday borrowers makes it clear that they are not a homogenous group. Some use payday loans out of convenience, some take out a payday loan because they have experienced an unexpected expense or financial shock, while others use them because they have no other options or because they are unsure if they qualify for mainstream credit. Although recent research examines the behavioral aspects of payday borrowing, and whether these loans improve borrowers' well-being (Caplan et al. 2017), there is still a gap in understanding regarding payday borrowers who experience difficulty in repaying loans. Our research contributes to understanding this issue by providing nuance to the gap.

We already know that there are obvious crisis borrowers, and this is confirmed by our research as well. Less has been said about the convenience borrowers who have no difficulty with repayment. However, our research clearly identifies an unsure group, and within that group, the lending community has an opportunity to provide innovative financial instruments and financial education to meet their needs. Primarily, they need a better understanding of their acceptability to conventional financial institutions and their personal credit information. They are apprehensive, as they perceive a lack of options to secure the funds that they need in a short period of time. It is this group for which we see a need for policy, targeted products, and financial education.

While economic models can provide useful predictions of the outcomes of decisions, they provide little "insight on the extent to which decisions are thoughtful and decisive" (Bolen et al. 2020, p. 1583). Using the theory of planned behavior (TPB) as a framework, we identify different consumer profiles of payday borrowers based on their attitudes towards payday lenders, perceived behavioral control, and subjective norms (positive, negative, or unsure). Using a questionnaire, we collected data from payday loan borrowers in a city with a population of over 1 million in a Western Canadian province. Then, using factor analysis and structural equation modeling, we examine how these characteristics are related to the difficulty in repaying loans, previous loan history, and their perceived quality of life after the loan. To our knowledge, we are the first to adapt the TPB framework to examine the experiences of payday borrowers. A number of variables have been used in the literature on borrowing, such as, confidence, attitudes, behavior, financial fragility, financial control, and well-being (Białowolski et al. 2020; Bialowolski et al. 2021), and we adapt this literature to the payday lending and borrowing market.

Based on our results of distinguishing attitudes and the perceptions of options based on borrowers' previous history with payday loans (difficulty in repaying or no difficulty in repaying), we argue that there is a unique opportunity to address needs based on borrowers' profiles. Different profiles have different needs regarding effective public policy, new credit products, and financial literacy. In addition, a better understanding of the different groups may reveal potential opportunities for collaboration among financial institutions—inside and outside of the mainstream—and provide solutions that meet the needs of different types of consumers. We provide an example of how a financial institution and payday lender recognized that every borrower is unique, and therefore collaborated to produce a product to better satisfy the needs of the payday borrower.

The rest of the manuscript is structured as follows. Initially, we set the stage for the research with the literature on the impact of payday loans on borrowers, the payday borrowers' consumer profiles, and the theory of planned behavior as adapted to the payday loan literature. Then, we provide the methods and materials used to investigate the research question. Following the methods, we analyze the questionnaire data and interpret the

findings. We conclude with implications for policy makers and examples of financial institutions successfully addressing distinct borrowing profiles, including opportunities to build financial confidence through the better targeting of financial literacy programs. Finally, we suggest fruitful directions for future research.

## 2. Related Literature and the Conceptual Framework

The literature on payday lending remains diverse yet inconclusive with respect to many questions (see Bolen et al. 2020). One stream of literature addresses whether the effects of payday lending on borrowers are positive, negative, or neutral. A second stream of literature examines how a change in regulations and policy affects the use of payday lending, and what effect the change has on borrowers. Because these two streams are interrelated, the major findings from both streams are discussed in the next section.

### 2.1. Impact of Payday Lending on Borrowers

Although Poschmann (2016) and Bolen et al. (2020) do not provide an exhaustive list of studies regarding payday loan use, both papers provide an overview of the diversity of findings regarding the social welfare benefits and costs of payday loans to the consumer (positive, negative, and no effect). Looking specifically at Poschmann (2016), of the 31 payday lending studies included in his review, ten studies were identified that found that payday loan use had positive effects for the borrower. Noteworthy were Morgan and Strain (2007), who examined two U.S. states that banned payday lending and found that rather than saving money, households paid substantially more in overdraft and returned check fees. As well, Morse (2011) found that an outright ban on payday lending increased the petty crime rate, and therefore concluded that payday lending contributed to the public good. In a similar vein, Bhutta et al. (2016) reported that when access to payday lending was restricted, consumers shifted to other forms of high interest credit instruments, such as pawnshops (Zinman 2010). Ellison and Forster (2008), although examining different markets and using different methodologies[1], found similar outcomes: that payday loans were beneficial. In some cases, access to payday loans promoted job retention (Zinman 2010) and financial well-being because borrowers missed fewer mainstream credit payments than those low-income credit users who chose credit card debt (Ellison and Forster 2008). In these instances, it would appear that accessing a payday loan was a rational decision made to assist the borrower to manage through difficult financial times. The more recent review by Bolen et al. (2020) reports similar positive effects.

Poschmann (2016) identified 12 studies that concluded that the existence of payday loans had a negative effect on borrowers, especially when regulation was inadequate. For example, Melzer (2011) found that payday loan access was associated with greater difficulty in paying expenses, and Carrell and Zinman (2014) found that it is related to poorer job performance. Skiba and Tobacman (2009) concluded that obtaining payday loan approval increased the likelihood of Chapter 13 bankruptcy, and Campbell et al. (2011) discovered that payday loan users had a higher likelihood of losing their bank accounts.

The last nine studies that Poschmann (2016) discussed found no impact on borrowers when payday loans were available. In this group, two noteworthy studies (Bhutta 2014; Bhutta et al. 2015) found evidence that the effects of payday borrowing on credit scores and other measures of financial well-being were close to zero (neutral).

The mixed results (positive, negative, or no effect) in the literature on the financial and welfare consequences of payday borrowing suggest that the answer to the question about the effect that payday loans have is ambiguous, and that the outcome may be dependent on specific borrower characteristics. Shapiro (2011) agrees. In his review of many of the studies contained in the Poschmann (2016) paper, Shapiro stated that "Virtually all of the studies, however, have significant methodological or data limitations which ultimately leave unresolved the central question of the net consumer effects of the loan" (Shapiro 2011, p. 1). Perhaps the answer is that payday loans are beneficial to some groups of payday

borrowers and not beneficial to others, depending on the consumer profile. We will discuss this topic next.

### 2.2. Consumer Profiles in Payday Borrowing

Many studies have attempted to identify and to consider both socio-economic characteristics and behavioral factors to facilitate a better understanding of borrowers, but very few have used survey methods to examine a borrower's decision processes for using small dollar credit products (Bolen et al. 2020). A number of studies provide a portrait of the average payday borrower. For example, Stango and Zinman (2009) found that payday borrowers typically had lower income and less education, did not shop for loans or alternatives, and were often white males. Bertrand and Morse (2011) identified low education level, frequent usage, the amount borrowed as a percentage of income, and inexperience in borrowing, as common characteristics of the typical payday borrower. However, MDRC (2016) found that typical assumptions around payday loan users (very low income, few assets, and low levels of education) did not hold for a large population of users. In their U.S. survey sample, 20 percent earned a net income of above USD 40,000 a year, about one-third were homeowners, and 40 percent of surveyed online subprime loan users had a four-year college degree or higher.

Other studies within the group explicitly identified different segments of payday borrowers based on a combination of demographic, socio-economic, and/or behavioral variables. Ranney and Cook (2011) identified four different non-prime consumer segments in the U.S.: emerging (no credit history), low income/credit disinterested, prior prime, and perpetually unstable. Each of these segments had distinct demographic and behavioral characteristics. For example, the emerging segment, which consisted of 1.1 percent of the population, included both (a) young consumers (between 18 and 25) with no credit and low salaries or hourly wages, and (b) immigrants who earned a low income and were yet to establish a solid credit history. The perpetually unstable consisted of 55.8 percent of the population. These borrowers appeared to be unable to plan for ordinary life events, facing a financial event every 25 days that they considered as beyond their control. The authors also found that consumers who defaulted on a payday loan exhibited distinctly different behaviors and attitudes from those who did not (e.g., more likely to rent, more likely to have changed checking accounts, and less likely to be aware of the interest rate paid). Their overall conclusion was " . . . consumers taking payday loans are largely everyday U.S. consumers who have a short-term need for cash but may not be able to get credit through the traditional channels. These consumers are self-identified as "generally financially healthy (50 percent)" (Ranney and Cook 2011, p. 3).

Another study by Dungen et al. (2016) classified payday borrowers into three groups: convenience-oriented, working poor, and hard to bank, using both socio-economic and behavioral factors; however, they provided little insight into why a payday loan might be chosen. Agarwal et al. (2009) attempted to identify reasons or offer explanations for why and when people used payday loans, even when the borrower had access to mainstream financial services or had substantial credit card liquidity. Similarly, Kobzar (2012a) offered categorizations based on behavioral explanations. She classified borrowers as: (1) credit addicted (a potential lack of financial literacy); (2) rational consumer (rational but credit constrained); and (3) bounded rationality (decisions impacted by circumstance, place, cultural, and ideological factors). She found that borrowers who exhibited "bounded rationality" chose payday loans as their best alternative. A small-scale exploratory study by Riley et al. (2022) that examined the lived experiences of a particular demographic of payday borrowers confirms many of Kobzar's findings.

The findings from the papers cited above (Ranney and Cook 2011; Agarwal et al. 2009; Kobzar 2012a) are consistent with an early study by Juster and Shay (1964), who developed a model of consumer borrowing decisions and considered the sensitivity to changes in interest rates. Contrary to the notion that payday borrowing is irrational, they proposed that payday borrowers might be "rationed" borrowers: borrowers who were constrained

by the creditors' equity requirements. The notion of the rationed borrower is consistent with later work. For example, using a cognitive model of decision making, Lawrence and Elliehausen (2008) found that nearly all payday borrowers were aware of the high finance charge, but not of the annual percentage rate (APR) charged. They surmised, as did Fusaro (2008), that customers weighed the cost of the payday loans relative to returned checks and late payment fees, making payday loans the best option available. Bolen et al.'s review (2020) found that pawns, vehicle title borrowers, and payday borrowers exhibit the characteristics of credit-constrained consumers.

Based on the research just discussed, and from the general borrowing literature, financial literacy is an important variable to consider when offering options for payday borrowers. However, there is also a general agreement that there is no standardized definition of financial literacy (e.g., Kimiyaghalam and Safari 2015; Ouachani et al. 2021; Zait and Bertea 2014). Noctor et al. (1992) made an early attempt to define financial literacy (as cited by Ouachani et al. 2021). According to the authors, financial literacy consists of two dimensions: the knowledge gained from some sort of education, and the ability to use the knowledge to make informed financial decisions. Other researchers have elaborated on this definition, expanding it to more dimensions.

From Kimiyaghalam and Safari's (2015) review of the literature, they found four concepts embedded in the definition of financial literacy: "(1) knowledge of financial concepts, (2) ability in managing personal finances, (3) skill in making financial decisions, and (4) confidence in future financial planning" (p. 81). Furthermore, Zait and Bertea (2014) proposed five main categories of financial literacy: knowledge, the ability to communicate that knowledge, the ability to use the knowledge to make decisions, the actual use of financial instruments (behavior), and confidence in the entire process. These definitions suggest that recalling terms, concepts, and calculations does not mean that individuals can actually apply their knowledge to make good decisions about their finances. In the next section, we explain how we address these dimensions in this research. Firstly, we focus on the behavior aspect of the financial literacy definition. Later, we link our findings to the financial knowledge aspect as we uncover aspects that affect the confidence of the borrower.

### 2.3. The Theory of Planned Behavior (TPB) in Consumer Financial Behaviors

We use the theory of planned behavior to investigate the behavioral aspects of payday borrowers, and then delve further into the "unsure" answers on our data collection instrument to suggest solutions for financial education that can improve the borrower's confidence.

According to the TPB, "[i]ntentions to perform behaviors of different kinds can be predicted with high accuracy from attitudes toward the behavior, perceived behavioral control, and subjective norms. Furthermore, these intentions . . . account for considerable variance in actual behavior" (Ajzen 1991, p. 179). The TPB has been shown to be valid in explaining behaviors and intentions related to financial decision making, such as credit card usage, student loans, and investing. The TPB suggests that to change behavior, the variables driving the behavior must be changed: specifically, attitudes, perceived behavioral control, and subjective norms. Similar to other researchers (e.g., see Godin and Kok 1996; Armitage and Conner 2001; Moss 2008), we adapt the TPB to reflect the specific considerations of our study and include important variables in borrowers' history that could affect the TPB variables.

Based on the work by Ranney and Cook (2011), who found that defaulting borrowers on payday loans exhibited distinctly different behaviors and attitudes from those not defaulting, we surmise that the theory of planned behavior (TPB) may be appropriate to delve further into why tension occurs in the literature as to whether payday loans provide a valuable service or lead to a cycle of debt. As mentioned earlier, the TPB has been applied to consumer financial behaviors such as student debt (Chudry et al. 2011; Xiao et al. 2011) and credit card use (Rutherford and DeVaney 2009). Rutherford and DeVaney (2009) found that

credit card holders who believe that credit is bad paid their credit card balances in full on a regular basis. Conversely, those with a positive attitude towards credit routinely paid only a portion of the balance and therefore accrued interest. Chudry et al. (2011) were among the first to use the TPB to examine the link between involvement with personal finance and actual behaviors. Student debtors responded to questions about their knowledge of sources of financing and their beliefs about the direct control they have over borrowing. The authors found that students considered themselves good money managers but perceived that they had little control over borrowing and debt to achieve their educational goals.

In the general borrowing literature, attitudes, behavior, knowledge, and confidence have also been identified as important variables in the financial literacy construct (Białowolski et al. 2020). We build on this literature but focus initially on the application aspect of the definition of financial literacy (capability or access and attitude) to understand why borrowers make the decisions that they do. Then, we use the findings to suggest options for improving their knowledge and skills, and thus their confidence. Rather than testing the general knowledge of the borrowing/lending process, such as interest rates, penalties, and other formal aspects that affect the borrowing decision, we ask the respondent about their more informal perceptions of their specific borrowing decision, such as the extent of their confidence in their lenders' characteristics and sources of borrowing available to them, and who influences their decisions. We build on both the TPB framework, the payday lending literature, and the general borrowing literature to offer insight into unique consumer profiles and suggest that each requires specific consideration to improve well-being and quality of life.

To investigate differences in consumers who use payday loans, we use the TPB because previous studies have shown its relevance to personal decisions regarding debt. These studies have found that the theory can be adapted to profile the market effectively based on consumer attitudes, perceived behavioral control, and subjective norms. Ajzen (1991) theorized that a person's behavior can be predicted through intention, and that intention is predicted by the individual's attitude toward the behavior, perceived behavioral control, and subjective norms. Later research by Romano and Netland (2008) suggested that if actual behavior is available, there is no need to capture intentions. Even though the model has been well researched and tested in a variety of applications, we refer to the TPB as a framework rather than a model in this exploratory research, as it has not been previously tested regarding payday borrowers.

As noted above, intentions can be predicted from attitude, perceived behavioral control, and subjective norms. According to the TPB, an individual's attitude toward a particular behavior is the "degree to which a person has a favorable or unfavorable evaluation or appraisal of the behavior in question" (Ajzen 1991, p. 188). In essence, this suggests a positive or negative evaluation of a particular behavior, based on that individual's beliefs. An individual's perceived behavioral control "refers to the perceived ease or difficulty of performing the behavior, and it is assumed to reflect past experience as well as anticipated impediments and obstacles" (Ajzen 1991, p. 188), or how difficult it will be to perform that behavior. The last factor is subjective norms. These are the perceived social pressures to perform or not to perform a certain behavior (e.g., what do those who are important to the individual think about that behavior). Subjective norms are important; yet, of the three factors, they are the most difficult to measure and assess. Ajzen (1991) found that in some instances, subjective norms show no clear discernable patterns. He explained the ambiguity of social norms by suggesting that in some situations, "personal considerations tended to overshadow the influence of perceived social pressure" (p. 189). Some researchers in the payday lending domain have recognized the challenge in measuring and assessing different aspects of these factors, such as overconfidence, the impact of prior financial education, and perceived credit constraints (see East 1993; Lyons 2004; Mandell and Schmid Klein 2009; Levinger et al. 2011; Bertrand and Morse 2011; Agarwal and Bos 2019; Harmon-Kizer 2019). In other words, the relative importance of the

three factors (attitude toward the behavior, perceived behavioral control, and subjective norms) varies across behaviors and situations.

*2.4. Extending the TPB Framework to Payday Borrowers*

As our research investigates payday borrowers (i.e., those who have taken out a payday loan) and not potential borrowers, we adapted the TPB to align with our research question. The limitation of the original TPB for our purposes is that it does not account for the factors that might influence how the attitudes, perceived behavior control, and subjective norms were formed initially. Ajzen (1991) also recognized the limitation of his original model and was open to including additional predictors that could further explain variances for which the original variables could not account.

Therefore, we followed the work of Xiao et al. (2011) and Chudry et al. (2011), who used external factors and past behavior, respectively, as antecedents to explain attitudes, perceived behavioral control, and subjective norms. Chudry et al. (2011) supported the use of past behavior in his extended TPB model to profile student debtors by citing several prior studies that found that previous behavior was a significant variable to be added to the TPB (e.g., see Marsh and Matheson 1983; Ouellette and Wood 1998).

Specifically, regarding payday loans and previous behavior, Ranney and Cook (2011) found that behaviors and attitudes were distinct between borrowers who defaulted and those who did not. Hence, we believe that this is an important variable for extending the TPB. Furthermore, previous behavior, such as having more than one payday loan at the same time, or using one loan to pay off another, could make it more likely that the borrower will have difficulty repaying the loan. This type of previous behavior often leads to a situation that is referred to in the general borrowing literature as financial fragility, or the inability to handle a small, unexpected expense. Generally, the greater the number of loans an individual carries, the greater the likelihood of repayment difficulty, and consequently, the fewer available choices for additional borrowing. It is often assumed that all payday borrowers are financially fragile and therefore enter a cycle of debt; however, our research findings and other studies suggest otherwise.

Ranney and Cook (2011) suggested that certain borrowers may not have access to traditional financial institutions, and therefore, they have little perceived behavioral control as to whether they use payday loans. Consequently, they make rational decisions based on what is available to them, and payday loans may be their best alternative (Kobzar 2012a; Riley et al. 2022). Both provide evidence that payday borrowing is a last chance alternative when borrowers find themselves facing an emergency financial situation. In contrast, Dungen et al. (2016), Caplan et al. (2017), and Bolen et al. (2020) recognize that borrowers fall into different classifications, such as those who use payday loans for convenience versus necessity. These three studies suggest that not all payday borrowers are facing a crisis financial situation and may use payday borrowing because it is quick and easy, even though they may have other borrowing alternatives.

Based on these findings regarding previous behavior, for our framework, we included antecedent variables to examine differences in attitudes and perceived behavioral control between payday borrowers who had difficulty repaying loans and those who did not have difficulty repaying their loans. This included examining whether the borrowers used one payday loan to repay another; as such, the behavior may lead to difficulties in repaying subsequent loans; therefore, we included a variable representing previous loans as well.

Regarding the consequences of the behavior in the model, we did not include intentions, but actual behavior outcome, similar to previous research that used actual behavioral data when available (e.g., see Godin and Kok 1996; Armitage and Conner 2001; Moss 2008). In our research, the outcome variable is the quality of life that the borrower experienced after the loan. The motivation for borrowers to take out payday loans is that they believe it will lead to better well-being, financially. Relying on the general literature on consumer borrowing, Bialowolski et al. (2021) provided extensive evidence from the extant literature that the borrowing process can have multiple outcomes regarding well-being, either neg-

ative or positive, including emotional, physical, and social well-being that are related to the borrower's quality of life after the loan decision. Testing both financial fragility and financial control, Bialowolski et al. (2021) found that financial control had a stronger size effect than financial fragility on emotional, physical, and partially on social well-being. However, the two variables, financial fragility and financial control, were not tested in the same model. We include a proxy both for fragility (using one loan to pay another) and for financial control (perceived behavioral control) in the same research, along with other relevant variables, for investigating the variability in the borrowers' perception of their quality of life after the loan. Due to the size of our sample, we did not attempt to distinguish between types of quality of life such as emotional, physical, and social. Our framework is illustrated in Figure 1. Each variable used is discussed individually, together with its related hypotheses.

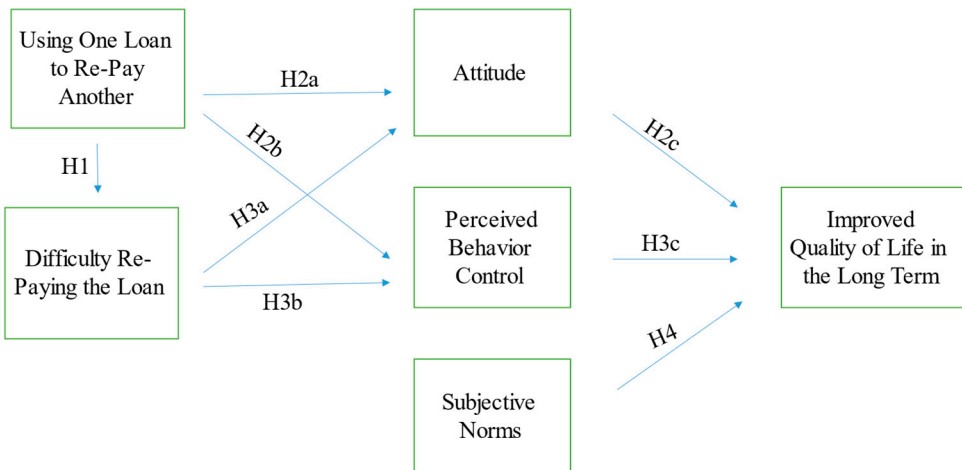

**Figure 1.** Model of Hypotheses.

### 2.5. Attitude

Consumer attitudes toward payday borrowing may stem from previous experience with payday lenders, advertising, or mainstream financial institutions. For example, a borrower has a significantly different experience with the loan application process of a payday lender, compared to a mainstream financial institution. Payday lenders ask for very little information[2], and typically within an hour, the applicant has the requested funds. Due to the convenience, it is not surprising that some individuals in financial need turn to payday lenders when fast cash is essential, particularly given the alternative of having to undergo the full application process required by a mainstream financial institution, which would likely include booking an appointment and consenting to a credit check (see Caplan et al. 2017; Bolen et al. 2020).

If borrowers have no difficulty repaying the loan and do not need to refinance their loan, this previous behavior may affirm their attitude that the fees and interest charged by payday lenders are reasonable for the quick and convenient service provided, and that payday lenders' business practices are fair. As well, some borrowers may believe that payday lenders provide better information and superior service compared to mainstream financial institutions, resulting in a positive attitude toward payday loans and lenders. If borrowers value convenience more than low cost or are willing to pay for that convenience, they are more likely to have a positive attitude toward payday loans and are less likely to default or to have difficulty in repaying the loan. This is consistent with the findings in Bolen et al.'s (2020) review, where "far more borrowers evaluated their decision to use payday loans positively, generally because the process provided needed funds quickly and conveniently (p. 1583)." Interestingly, Riley et al. (2022) identified loyalty to the payday lender as a potential influence. That is, having a positive relationship with the lender

helped to hide the embarrassment of using the product, because only a few people would know that the borrower needed assistance.

However, negative attitudes may be formed regarding the fairness of the payday lenders if the borrower is unable to repay their loans when they come due or need to refinance their loans by taking out other loans. That is, the inability to repay may cause the borrower to form a negative attitude toward the lender's service, the lender's ability to understand the borrower's needs, and the justice of the process. Bolen et al. (2020) found that of the dissatisfied small dollar borrowers, half indicated concern about the price of the loan, and 17 percent indicated concern about their ability to get out of debt; however, the 17 percent of those dissatisfied represented only 3 percent of all borrowers.

The differences in the characteristics of these borrowing types lead to our first set of hypotheses. Based on the extant literature regarding previous experiences influencing attitudes, we first test whether using a loan to repay a previous loan leads to difficulty in repaying the latest payday loan. Even though this may seem obvious, the hypothesis is necessary to establish a difference between the borrowers who are caught in a cycle of debt and those borrowers who use payday borrowing for convenience. Next, we test if difficulty in repaying the loan influences the attitude that the borrower has toward the payday lender. Finally, we test if the previous loan and the difficulty repaying the loan influences the borrower's quality of life following the receipt of the loan.

**H1.** *The payday borrower who used (did not use) one loan to repay another loan is more likely to have (not have) difficulty repaying a loan. (Direct effect.)*

**H2a.** *The payday borrower who used (did not use) one loan to repay another loan is more likely to have a negative (positive) attitude toward the service of payday lenders. (Direct effect.)*

**H2b.** *The payday borrower who had (did not have) difficulty repaying the loan is more likely to have a negative (positive) attitude toward the service of payday lenders. (Direct effect.)*

**H2c.** *The payday borrower who had a negative (positive) attitude toward the service of payday lenders, due to using one loan to repay another loan and/or difficulty repaying the loan, is more likely to have a negative (positive) effect on quality of life following the loan. (Direct and indirect effects.)*

*2.6. Perceived Behavioral Control*

Studies that attempted to identify different consumer profiles of payday borrowers all described a group that uses payday loans in a rational way: to access cash quickly to meet a temporary shortfall, without the time and effort required by mainstream financial institutions (Birkenmaier and Fu 2016). However, there are also those financially fragile payday borrowers who, due to the combination of their inability to meet unexpected expenses and their limited access to other credit sources, use payday loans because they have no other options (Riley et al. 2022). Zinman (2010) found that 70 percent of survey respondents reported that there were few options for borrowing outside of payday loans, and Ellison and Forster (2008) found that one-third of payday borrowers indicated that payday loans were the only source of credit available to them. Bhutta et al. (2016) found nearly 21 percent of respondents indicated that payday loans were used because traditional banks did not provide small loans.

The preceding discussion indicates that some individuals perceive that few alternatives exist for accessing small funds quickly (e.g., under USD 1500). However, appropriate financial education might change this perception, as some mainstream financial institutions are attempting to accommodate the needs of payday borrowers by offering unique credit products. Kobzar (2012a) provides anecdotal evidence that some customers felt a sense of exclusion from mainstream financial institutions due to the potential (whether real or perceived) for their loan application to be denied due to poor credit history. Agarwal and Bos (2019) conducted an empirical study on the use of pawnbroker loans in Sweden and found that borrowers who did not attempt to find cheaper mainstream credit before

obtaining a pawnbroker loan had roughly the same probability of having their credit application approved as the median-granted group in the general population. Kobzar (2012a) surmised that this feeling could be exacerbated by the payday loan industry, given that some respondents reported that certain payday lenders communicated to borrowers that the cost of mainstream credit was exorbitant, and that banks were only for the rich. Levinger et al. (2011) examined the impact of self-perception on making suboptimal credit decisions. They found strong evidence that an applicant's perceived odds of being approved impacted on the decision to apply for the loan and the applicant's self-assessment of their creditworthiness. Thus, borrowers that have difficulty repaying their loans and who perceive they do not have other alternatives, or they actually do not have other alternatives, are more likely to determine that they are locked in and cannot control their own behavior. These perceptions offer a unique opportunity for building financial confidence through financial education.

On the other hand, payday borrowers who have had no previous experience with default may perceive that if they were successful with payday loans in the past, they could also be successful with mainstream financial institutions but still choose the payday loan process (Ranney and Cook 2011; Dungen et al. 2016). They may have an acceptable credit rating and be welcomed at traditional banks and credit unions, but they choose the payday lender for other reasons of privacy, timeliness, a lack of financial literacy, or convenience, and are willing to pay a higher interest rate for these more intangible benefits (Agarwal et al. 2009; Kobzar 2012a).

Further advancing the argument that previous behavior with payday loans is an important variable which also influences the perceived behavioral control of the borrower, we propose the following three hypotheses.

**H3a.** *The payday borrower who used (did not use) one loan to repay another loan is more likely to have negative (positive) perceived behavioral control. (Direct effect.)*

**H3b.** *The payday borrower who had (did not have) difficulty repaying a loan is more likely to have a negative (positive) perception of behavioral control. (Direct effect.)*

**H3c.** *The payday borrower who had negative (positive) behavioral control, due to using one loan to repay another and/or difficulty repaying the loan, is more likely to have a negative (positive) effect on quality of life following the loan. (Direct and indirect effects.)*

### 2.7. Subjective Norms

According to the TPB, an individual's behavior is affected by subjective norms. These norms are formed based on a person's beliefs about whether people important to them support, affirm, or discourage the behavior. If people important to an individual encourage the behavior, there is an expectation that the individual is more likely to engage in that behavior.[3] For example, Chudry et al. (2011) found that although student debtors were well aware of their parents' concerns about debt, students potentially expressed their independence by taking on debt despite their parents' warnings, and the norms were found to be insignificant in predicting students' debt behavior. A similar result was found in Xiao et al. (2011). In their adapted TPB model, they found that friends' norms were not significant in preventing students from building up credit card debt.

Regarding payday borrowers, not much is known about their decision-making process when it comes to the influences of family members and friends. Mandell and Schmid Klein (2009), Friedline and Kepple (2017), and Harmon-Kizer (2019) examined different aspects of influence (family, friends, and social structural mechanisms). In general, they found that financial behaviors might be influenced by subjective norms, but none were able to capture the extent of the influence. It might be that friends and relatives are a payday borrower's primary source for help and advice for a payday borrower, or it might be that money matters are private and are not discussed with family or friends. Therefore, one could argue that if the norm were to use payday loans, and that a family member or friend

had a good experience or even no experience, the potential borrower might be influenced and assume that taking out a payday loan would lead to a better quality of life. Therefore, we only predict that knowing a family member or a friend who had experience with payday loans would influence the borrowers' perception about the loan leading to a better quality of life.

**H4.** *The payday loan borrower who knew someone who had (had not) used a payday loan is likely to believe that it will lead to a positive (negative) effect on quality of life. (Direct effect.)*

### 3. Methods and Materials

Data for our study were collected through a citywide committee called the Safe and Affordable Financial Instruments Committee in a western province in Canada. The purpose of the study was to shed light on the payday lending market to make pending changes in policy and regulations relevant to different borrowers' characteristics. However, the Committee's primary objective was to identify borrowers who were having difficulty repaying the principal, interest, and fees on their payday loans within the terms. The Committee wanted to determine the characteristics that lead to the financial troubles associated with payday loans to develop alternative products that were not currently in the market. Consequently, the targeted respondents for the survey were payday loan borrowers, not the general population.

To develop the questionnaire, we first reviewed the payday loan literature to identify borrower characteristics that were relevant to our research question. Using the TPB, we developed a set of questions relating to attitudes, perceived behavior control, and norms for payday loan borrowers. To gather previous behavior, the survey also included questions pertaining to any difficulties of repaying the loan, using one loan to pay off another, and loan amounts, which measures financial fragility. Regarding outcomes, questions pertained to quality of life and improved financial situation. We relied on the general literature about the effect of loan behavior on well-being. Furthermore, general demographic and socio-economic questions regarding gender, age, ethnicity, marital status, employment status, frequency of paychecks, household income, and loan history were used. The Committee reviewed the survey to ensure that the questions were relevant to their needs. Respondents answered the questions pertaining to the TPB using a 5-point Likert scale, ranging from 1 (strongly disagree) to 5 (strongly agree). Respondents were also given a possibility of answering with "unsure."

Committee members associated with social service agencies made the questionnaires available at their agencies. Some of them asked their clients, when visiting for services, whether they had a payday loan in the past or currently had one. If they answered "yes" they were asked to complete the questionnaire. One payday lender also made it available in his place of business and asked his clients to complete it. All respondents were offered a gift card of CAD 5 to use at a well-known coffee shop. Answers were reviewed for completeness, resulting in a final sample of 138 respondents. See Table 1 for the breakdown of respondents from different types of organizations.

**Table 1.** Percentage of Samples from each Type of Organization.

| Type of Organization | Percentage of Respondents |
|---|---|
| Addiction Centers | 20.7 |
| Micro-Finance Organizations | 17.9 |
| Payday Lender Establishments | 11.7 |
| Emergency Shelters | 22.8 |
| Various Social Service Agencies | 26.9 |
| Total | 100.0 |

Univariate analysis was used to report the results of the demographic analysis. Factor analysis and structural equation modeling were used to analyze the results of the survey.

Factor analysis was used to test the validity of the questionnaire items based on the TPB variables of attitude, perceived behavioral control, and subjective norms. Structural equation modeling (SEM), which consists of a series of regression equations, was used to test the variables associated with each hypothesis, including the variables representing the respondent's history, TPB variables, and the outcome variable.

## 4. Results

### 4.1. Univariate Analysis

The final sample consisted of 138 surveys. The sample's demographic characteristics are presented in Table 2. Our sample of respondents consisted of 63 percent male, 35 percent female, and 2 percent other. Respondents' ages ranged from 18 to over 65, with the range of 25 to 35 and 35 to 44 both consisting of 26 percent. Over 66 percent of the sample was Caucasian, and 20 percent were Aboriginal. Most of the sample was single (81 percent). Given that a requirement to receive a payday loan is employment, it might appear surprising that more than half (56 percent) of the sample was unemployed. However, this is likely due to a time lag between the date of the payday loan and when the question was answered, for some respondents. Hence, respondents could have been employed at the time of the loan but unemployed when the question was answered.

**Table 2.** Demographic Characteristics of the Sample of Payday Borrowers (n = 138).

| Gender | Male | Female | Other | | | |
|---|---|---|---|---|---|---|
| Percentage | 63% | 35% | 2% | | | |
| Age | 18–24 | 25–34 | 35–44 | 45–54 | 55–64 | 65+ |
| Percentage | 9% | 18% | 26% | 26% | 18% | 2% |
| Ethnicity | Caucasian | Latino | Black or African | Aboriginal | Asian | Other |
| Percentage | 66% | 0% | 2% | 20% | 3% | 9% |
| Marital Status | Single | Married/ Common Law | | | | |
| Percentage | 81% | 19% | | | | |
| Employment | Working for others | Self-employed | Unemployed | | | |
| Percentage | 39% | 5% | 56% | | | |
| Paycheck Frequency | Weekly | Every two weeks | Monthly | Other | | |
| Percentage | 6% | 17% | 37% | 28% | | |
| Income | Under CAD 19,999 | CAD 20,000– 39,999 | CAD 40,000– 59,000 | CAD 60,000– 79,0000 | CAD 80,000– 99,000 | CAD 100,000– 119,000 | CAD 120,000+ |
| Percentage | 53% | 27% | 15% | 2% | 0% | 2% | 2% |
| Loan Frequency | Within last 6 months | Within last year | Within last 2 years | More than 2 years ago | Cannot remember | |
| Percentage | 29% | 26% | 9% | 27% | 9% | |

In our sample, 37 percent received a paycheck monthly, and the income ranged from under CAD 19,000 (53 percent) to over CAD 120,000 (2 percent). In terms of loan frequency, the sample had a high frequency of use for those using payday loans within the last year and more than two years ago: 29 percent had a loan within the last six months, 26 percent within the last year, only 9 percent within the last two years, and 27 percent more than two

years ago. Loan amounts ranged from CAD 100 to CAD 8000, with a mean of CAD 602 and a median of CAD 350 (not tabulated).

Whether payday borrowers had difficulty repaying their loans was a primary focus of our study. Out of the respondents, 63 percent (83 respondents) experienced difficulty repaying a loan, and 37 percent (51 respondents) did not. There were no significant differences in age, gender, ethnicity, marital status, paycheck periodicity, household income, loan frequency, or loan amount between those who had difficulty and those who did not have difficulty repaying their loan. Employment was the one variable that was significantly different between borrowers who had difficulty repaying a loan and those that did not. As well, 38 percent (52 respondents) answered "yes" to using one payday loan to pay off another, with 62 percent (84 respondents) answering "no." Two respondents did not answer this question.

### 4.2. Factor Analysis

The 12 questions designed to assess attitudes, perceived behavioral controls, and norms were entered into a factor analysis, three-factor solution. The matrix was rotated using a Varimax with Kaiser Normalization rotation. Three unique factors were extracted, each with eigenvalues of greater than 1. Together they explained 60 percent of the variance. The three factors individually explained 30 percent, 21 percent, and 9 percent (rounded) of the total variance in the responses. The rotated component matrix converged in four iterations and is shown in Table 3.

**Table 3.** Factor Analysis: Rotated Component Matrix [a].

| Item | Component Factors | | |
|---|---|---|---|
| | **Attitudes** | **Perceived Behavioral Control** | **Subjective Norms** |
| The business practices of payday lenders are fair. | **0.853** | −0.024 | 0.042 |
| I trust that my payday lender will provide me with superior service. | **0.835** | 0.045 | 0.079 |
| Payday lenders charge reasonable fees and interest. | **0.770** | −0.120 | 0.086 |
| I trust that my payday lender will always provide me with correct information regarding the loan process. | **0.731** | 0.037 | 0.205 |
| My payday lender understands my money needs. | **0.622** | −0.044 | 0.398 |
| Banks and credit unions would likely reject me if I asked for credit. | −0.029 | **0.859** | 0.007 |
| I use payday lending because my credit rating is not good enough to borrow from a bank or credit union. | −0.037 | **0.804** | 0.099 |
| Banks and credit unions do not welcome people such as me. | 0.091 | **0.748** | −0.027 |
| I only use payday loans because I have no other options. | −0.145 | **0.683** | 0.316 |
| I know someone who has received a payday loan. | 0.029 | 0.184 | **0.828** |
| People who are important to me have encouraged me to take out a payday loan. | 0.260 | −0.018 | **0.596** |
| My relatives have used payday loans. | 0.137 | 0.084 | **0.588** |

Extraction Method: Principal Component Analysis Rotation Method: Varimax with Kaiser Normalization [a]; a. Rotation converged in 4 iterations. The high loadings for each factor are bolded, indicating the three distinct factors.

None of the factors had high cross-loadings, and the Cronbach alphas were as follows: Factor 1 containing five items representing attitude with a Cronbach's Alpha of 0.844, Factor 2 containing four items representing perceived behavioral control with a Cronbach's Alpha of 0.791, and Factor 3 containing three items representing subjective norms with a Cronbach's Alpha of 0.540. Similar to previous studies, the subjective norms factor exhibits low reliability.

Based on the previous literature, we hypothesize that there are different profiles for payday borrowers. After analyzing the characteristics that are associated with each identified group, we refer to these two groups as the crisis group (those who have difficulty repaying the loan) and the convenience group (those who did not have difficulty repaying the loan). We ran *t*-tests on the items included in each factor to determine if the two groups have different attitudes, perceived behavioral control, and subjective norms. Based on the Likert scale raw scores, the mean and level of significance for each of the items in each of the factors is shown in Table 4. The results show a significant difference ($p \leq 0.01$) among the attitude and perceived behavior control items used in these two factors, but not among the subjective norms items used in the third factor.

**Table 4.** *t*-test on Factor Items for Difficulty (Crisis) and No Difficulty (Convenience) Repaying the Loan.

| Item | Significance Level | Mean: Difficulty Repaying N = 87 | Mean: No Difficulty Repaying N = 51 |
|---|---|---|---|
| **Attitude** | | | |
| Fees reasonable | *** | 1.75 | 2.92 |
| Correct information | *** | 2.89 | 3.48 |
| Fair practices | *** | 2.20 | 2.98 |
| Superior service | *** | 2.46 | 3.46 |
| **Behavioral Control** | | | |
| Bank and credit unions do not welcome | *** | 3.57 | 2.98 |
| Banks and credit unions likely reject | *** | 4.07 | 3.30 |
| No other options | *** | 4.26 | 3.38 |
| Credit rating not good enough | *** | 3.86 | 3.06 |
| **Norms** | | | |
| People important to me encouraged a payday loan | NS | 2.36 | 2.28 |
| Knows someone who received a payday loan | NS | 4.26 | 4.00 |
| Relatives use payday loans | NS | 3.30 | 3.02 |

*** $p = 0.01$; NS = Not significant. Note: *t*-test results show significance on responses of difficulty and no difficulty repaying. Attitude and Behavioral Control statements are all significantly different between those having difficulty repaying a loan and those not having difficulty repaying the loan ($p = 0.01$). Norm statements are not significantly different between the two groups.

### 4.3. Structural Equation Modeling (SEM) Analysis

To test the hypotheses and the fit of the model, we used structural equation modeling (SEM) analysis. Overall, the independent variables explain 10 percent of the variances of using one loan to pay off another, 16 percent of the variance in attitude toward the loan, 21 percent of perceived behavioral control, and 58 percent of the improved quality of life in the long run. See Figure 2 for the results of the hypotheses testing, which are discussed in more detail later in this section.

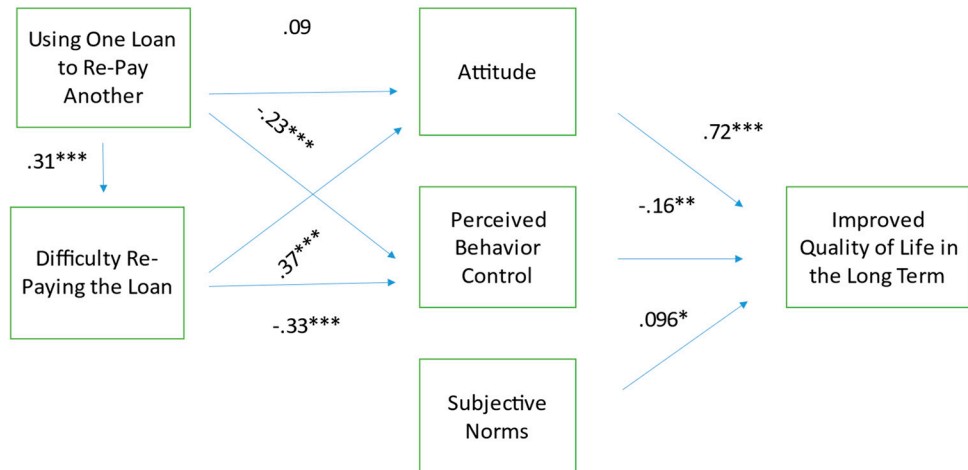

**Figure 2.** SEM Model of Hypotheses Testing. ***, ** and * denote significance levels (two-tailed) at 1%, 5% and 10%, respectively.

Although the fit indices may not be valid, given our small sample size, we provide the results of the most common fit indices compared to the common benchmarks. Fit indices compare our suggested model with the saturated[4] model (Hampton 2015). The fit indices, including the likelihood ratio and chi-squared value, indicate that the model is well specified (Kline 2015). The null hypothesis is not rejected, which means that the model is adequate in comparison to the saturated model (full model) (chi$^2$ = 13.707, $p$ > 0.033). The insignificance of chi$^2$ implies that there is not a missing path in the model. The summary statistic does not reject its null, which is an indication of model validity.

Moreover, the comparative fit index (CFI) and Tucker-Lewis index (TLI) statistics of 0.957 and 0.899 are near the suggested level of 0.96 and 0.95, respectively. The standardized root mean squared residual (SRMR) of 0.057 is less than the suggested threshold of 0.09. The root mean square error of approximation (RMSEA) of 0.099 is near the suggested value of 0.08. All of these fit indices indicate that the model provides a fairly good fit to the observed data (Hu and Bentler 1999; Kline 2015; Tabachnick and Fidell 2019); however, we should keep in mind that our sample size is small, which may affect the validity of the fit indices.

The results of the SEM model, including the total, direct, and indirect effects of the variables of interest (using one loan to pay off another, difficulty repaying the loan, attitude toward payday loan, perceived behavioral control, subjective norms, and quality of life in the long term) are illustrated in Table 5. The results show that using one loan to pay off another loan is significantly related ($\beta$ = 0.31, $p$ < 0.01) to the difficulty that a borrower has in repaying the payday loan. This supports the significance of financial fragility, or the inability to handle a small, emergency expense, as leading to a higher probability of default on the payday loan, supporting H1. However, it also provides evidence that not all payday borrowers are financially fragile or enter a cycle of debt.

Concerning the attitude that borrowers have about payday lenders, whether a borrower had difficulty in repaying the loan can significantly influence the attitude ($\beta$ = 0.39, $p$ < 0.01). Difficulty in repaying a loan leads to the feeling that payday lenders are unfair, that they do not provide good service, and that their fees are unreasonable. Furthermore, the borrowers believe that they do not receive correct information from the lender, and that the lender does not understand their financial needs. In contrast, not experiencing difficulty in repaying the loan is associated with attitudes of fairness, superior service, reasonable fees, correct information, and a good understanding of the borrowers' financial needs. Using one loan to pay off another loan does not have a significant effect on attitude; rather this variable works through the difficulty repaying variable to create a negative or positive attitude.

**Table 5.** Goodness-of-Fit Indices for the Structural Equation Model.

| | |
|---|---|
| **Equation Level R-Squared** | |
| Use one Loan to Repay Another | 10% |
| Attitude Toward Payday Loans | 16% |
| Perceived Behavioral Control | 21% |
| Improved Quality of Life in the Long Run | 58% |
| Overall | 30% |
| **Fit Statistic Likelihood Ratio** | |
| chi2_ms(6) | 13.707 |
| $p > \text{chi}^2$ | 0.033 |
| **Population Error** | |
| RMSEA | 0.099 |
| **Baseline Comparison** | |
| CFI | 0.957 |
| TLI | 0.899 |
| **Size of Residuals** | |
| SRMR | 0.057 |
| CD | 0.300 |

Regarding perceived behavioral control, both variables, using one loan to pay off another ($\beta = -0.23$, $p < 0.01$) and difficulty repaying the payday loan ($\beta = -0.39$ $p < 0.01$), will influence the perceived behavioral control of the borrower or the ability to control their borrowing options. However, the difficulty repaying variable has a stronger effect on the perceived behavioral control than the variable of using one loan to pay off another loan. Those who had difficulty in repaying the loan perceived that banks and credit unions would not welcome them and would likely reject them, that they had no other options, and that their credit rating was not good enough. In contrast, those who did not have difficulty repaying the loans responded in disagreement with these same statements.

Proceeding to the next level of the model, we now discuss the extent that the borrowers felt that the loan had on improving their quality of life. Difficulty repaying the payday loan ($\beta = 0.35$, $p < 0.01$), attitude toward payday loans ($\beta = 0.72$, $p < 0.01$), and perceived behavior control ($\beta = -0.16$, $p < 0.05$) show a significant effect on the borrowers' perceptions that the loans improved the quality of their lives in the long run. In other words, if the borrower had no difficulty repaying the loan, the borrower's attitude was that the lender's service was good and that the borrower had other options; therefore, the loan improved the borrower's quality of life. In contrast, if the borrower had difficulty repaying the loan, the borrower's attitude was that the lender's service was not good, and that the borrower had no other options; therefore, the quality of life was not improved in the long run. The subjective norms factor ($\beta = 0.10$, $p < 0.1$) is marginally significant on the improved quality of life in long run for the payday borrower. If the borrower knew others who had payday loans, the borrower felt that the loan improved their quality of life.

In sum, respondents who did not experience difficulty in repaying the loan had more positive attitudes toward payday lenders and were more likely to perceive that the loan improved their quality of life. In contrast, if the respondent did experience difficulty in repaying the loan, the borrowers' attitudes toward payday lenders were more negative, as was the borrower's assessment of quality of life, providing support for Hypotheses 2a, 2b, and 2c. Furthermore, those borrowers who had difficulty repaying the loan also perceived a low level of behavioral control (i.e., a lack of options for borrowing or control of financial alternatives) and that the loan did not improve their quality of life. In contrast, those who had no difficulty repaying the loan also perceived a high level of behavioral control and an improved quality of life, providing support for Hypotheses H3a, H3b, and H3c. Finally, those who used one loan to pay another loan were more likely to experience difficulty repaying the loan, also affecting the attitude and behavioral control variables. There was marginal support for H4, that borrowers who knew someone who had used

payday loans believed that they would experience a better quality of life. These results support the importance of financial fragility on the ability to repay a payday loan. It also provides strong support for profiling payday loan borrowers into convenience and crisis groups, rather than treating all borrowers as the same. More clearly defining characteristics of the groups having difficulty repaying the loans helps to understand how to serve these groups through targeted education and specialized financial instruments.

### 4.4. Further Profile Analysis: The Unsure Group

The previous SEM analysis used the entire sample to examine the role of attitudes, behavioral control, and norms related to the difficulty in repaying a payday loan to characterize two groups of payday borrowers, termed convenience and crisis. To gain additional insight into the potential to educate payday borrowers of their best options, we delved deeper into those respondents answering "unsure" to the factors used in the analysis in both of these groups. This additional investigation provided insight into the financial confidence of their borrowing process and their options. We conducted further analysis to understand the experience of this group with payday loans, given that some had difficulty repaying and others did not. Please see Table 6.

**Table 6.** Percentages "Unsure" on Factor Items for Difficulty (Crisis) and No Difficulty (Convenience) Repaying the Loan.

| Item | Percentage of Unsure for Difficulty Repaying | Percentage Unsure for No Difficulty Repaying |
|---|:---:|:---:|
| **Attitude** | | |
| Fees reasonable | 6 | 8 |
| Correct information | 23 | 16 |
| Fair practices | 13 | 28 |
| Superior service | 36 | 24 |
| **Behavioral Control** | | |
| Bank and credit unions do not welcome | 35 | 34 |
| Banks and credit unions likely reject | 24 | 30 |
| No other options | 8 | 20 |
| Credit rating not good enough | 22 | 10 |
| **Norms** | | |
| People important to me encouraged a payday loan | 11 | 20 |
| Know someone who received a payday loan | 5 | 6 |
| Relatives use payday loans | 18 | 26 |

For the items comprising our two main factors (behavioral control and attitude), we found that a relatively high percentage of respondents answered "unsure" to a number of questions, indicating a lack of financial confidence and a need for financial education. The unsure group represents a sub-group of each of the crisis and convenience groups. For the items comprising attitude, borrowers answered "unsure" relatively often: regarding whether lenders provide correct information (23 and 16 percent: difficulty and no difficulty repaying, respectively), have fair practices (13 and 28 percent), and provide superior service (36 and 24 percent). The fact that a significant portion of payday borrowers—particularly those who had difficulty paying back their loan—are uncertain about whether lenders provide correct information, have fair practices, and provide superior service is consistent with the notion that these borrowers do not have other options (or that they do not believe they have other options). As a result, they rely on payday loans despite being skeptical of the product/practices.

Regarding the availability of alternatives to payday loans (behavioral control), it appears that borrowers in both groups were approximately equally concerned with approaching a mainstream financial institution and feeling unwelcomed or rejected. For example, 35 percent of those having difficulty repaying the loan and 34 percent (approximately a third of each group) of those not having difficulty repaying the loan were unsure about being welcomed. However, between these two groups, there were significant differences in the awareness of other alternatives and credit ratings. For example, for those experiencing no difficulty in repaying the loan, 20 percent were unsure about other options, and 10 percent were unsure about their credit rating. Of those who had difficulty repaying, 8 percent were unsure about other options, and 22 percent were unsure about their credit rating.

This indicates that some borrowers can choose the convenience of payday loans over the effort and time required to borrow from a mainstream financial institution. Further, it suggests that there is a sub-group of payday borrowers that may qualify for the services of mainstream financial institutions, and yet they may lack the knowledge needed to navigate the system to investigate other alternatives, and/or lack the time or willingness to do so. The identification of a sub-group of payday borrowers who are "unsure" about other options that are available to them, their credit scores, and whether they qualify for mainstream credit is an important contribution to the understanding of payday borrowers. Our results regarding "convenience" and "crisis" borrowers further support earlier work that shows that payday loans are used by different groups for different reasons. While the convenience borrowers appear to be making a rational choice, the crisis borrowers more likely should not be using payday loans at all. However, when faced with an emergency situation, they may be making the best choice available to them.

The unsure sub-group, identified in this research, consists of borrowers who are the least understood. Given the social and individual costs associated with payday loans when borrowers have difficulty repaying, we review and provide a discussion about some of the recent public policies that have been proposed to help address this sub-group. We see this group benefiting primarily from carefully designed educational products, transparent promotion, and financial products that meet their unique needs.

## 5. Discussion and Public Policy Implications

Payday loans are a controversial practice and research addressing the impact of payday borrowing on users is mixed. Whether borrowers are prey or purchaser, in crisis or using payday loans for convenience, and creditworthy or unsure if they qualify for credit, are important facts to know regarding payday borrowers. However, the perspective taken by a payday borrower is complicated and complex. Understanding how changes in public policy can influence borrower behaviors is equally difficult given the numerous interrelated factors affecting the supply and demand side of payday loans. The payday lending topic, regardless of how it is perceived, cannot be tackled using simple rules and processes.

To our knowledge, the TPB has not been used previously to study payday loans, making this a unique contribution to the literature. Using results from a questionnaire sent to payday borrowers and the TPB framework as a case study for the payday lending situation in one city, we found unique characteristics among payday borrowers. We classify them into three groups: crisis, convenience, and unsure, with the unsure group offering the greatest opportunity for improvement through policy, instrument innovation, and targeted financial education.

We examined how using one loan to pay back another loan creates financial fragility and affects the difficulty in repaying a loan, confirming the crisis group, which is also supported in past literature. Furthermore, the difficulty that the borrower has in repaying a loan affects the attitudes and perceived behavioral control, and the associated perception of the improvement in quality of life that the loan provided. Subjective norms had only a marginal significance on the perception of quality-of-life improvement. However, we also found that the crisis group is not a valid descriptor for all payday borrowers. Some

borrowers use payday loans for convenience, with few or no detrimental effects. This group has positive attitudes toward payday borrowers and does not become caught up in a cycle of debt, providing them with an improved quality of life. Finally, the unsure group lacks financial confidence and exists as a sub-group of both the crisis and convenience groups due to a lack of understanding about financial institutions' offerings and their own personal credit situation. This group provides a clear opportunity to move away from payday loans if conventional banks are willing to offer innovative financial instruments to accommodate this group, and for financial literacy institutions to ensure that the unsure group knows that other alternatives are available to them.

Previous studies (e.g., Ranney and Cook 2011; Kobzar 2012a; Caplan et al. 2017; Caplan et al. 2017; and Bolen et al. 2020) have recognized that payday borrowers are not a homogeneous group, suggesting the importance of profiling them to understand the characteristics of each group when developing appropriate policy. Building on previous studies (Chudry et al. 2011; Xiao et al. 2011; Rutherford and DeVaney 2009), we found support for two of our three hypotheses: payday loan borrowers who have difficulty repaying their loans have lower levels of perceived behavioral control and negative attitudes toward payday lenders, which relates to their perception of whether the loan improved their quality of life. Their borrowing history also plays a role in their ability to repay the loan. Subjective norms were marginally significant regarding the loan's ability to improve the borrowers' quality of life.

Although our findings align with some previous studies on payday lending that did not differentiate the market, we believe that the identification of market profiles may help to explain some of the variances in the findings of those studies. For example, Zinman (2010), Ellison and Forster (2008), and Bhutta et al. (2016) all identified that some percentage (varying from as high as 70 percent to as low as 21 percent) of payday borrowers have no or few options when in need of short-term cash, and therefore they must resort to payday lenders. When we group the borrowers into crisis and convenience categories, we found that the crisis group expressed a greater concern about a lack of options than the convenience group, likely due to their loan history of having more than one payday loan at the same time. Payday borrowers who use loans for convenience are not a primary concern if they do not have difficulty repaying the loan. However, for public policy purposes, it is essential to have a better understanding of the group that has difficulty in repaying loans. The unsure sub-group, within both the crisis and convenience groups, each have specific characteristics and may be underserved by the current menu of options available to meet short-term cash needs. Financial literacy is usually the first solution proposed by those working to wean borrowers from payday loan usage. Rather than using financial literacy as a lone solution, we believe that a better approach is to complement financial literacy efforts with the understanding that for some borrowers, the lack of alternatives (real or perceived) result in their use of payday loans. For example, the mixed results on the benefits of financial education from previous studies (see Haynes-Bordas et al. 2008; Mandell and Schmid Klein 2009; Bertrand and Morse 2011; Friedline and Kepple 2017, and others) suggests that it might be useful for financial education programs to specifically include information designed to improve awareness and accessibility to credit scores, and information on how scores impact borrowing.

The crisis/unsure group is a market that most mainstream financial institutions are hesitant to service, given the small dollar amounts of the potential loans, the high default rates, and the administrative cost required to process applications. As well, payday lenders are unlikely to perceive the crisis group as a profitable business group. Financial institutions (alone) may not be motivated to service this group on their own, but organizations such as the United Way and micro-credit establishments have suggested that cooperation between stakeholders could provide a viable solution. Nevertheless, several credit unions in the location of the study have developed unique products that address the need for quick funds at reduced interest rates to offer options to the payday borrower. Ideally, social agencies, regulators, and policy makers can work together to create solutions for this group, rather than expecting payday borrowers to change their behavior in the absence of appro-

priate alternatives. Such a call for collaboration is not new; a number of researchers have made similar recommendations (for example, see Birkenmaier and Fu 2016; Friedline and Kepple 2017). However, policy changes without addressing the need for alternative credit instruments and financial education can make the situation worse for payday borrowers.

While competition and consumer credit markets have changed substantially over the past number of years, several progressive credit unions and provincial financial institutions in Canada recognize the need to take a different approach. For example, ATB (Alberta Treasury Branch) is a provincial financial institution that provides a full range of financial services. ATB joined forces with Cashco, a payday lender, to capitalize on the quick and convenient service provided by payday lenders, and the financial products offered by mainstream financial institutions. Through the ATB/Cashco agreement, payday loan customers have access to overdraft-protected ATB deposit and transaction accounts. With the financial backing of ATB, Cashco can now offer its customers loans with reduced interest rates (Strader 2017). ATB also offers Cashco customers access to financial literacy education and personal budgeting to provide crisis/unsure customers an opportunity to use mainstream financial services through their payday lenders without fear of rejection. If the payday borrower then becomes a customer of ATB, then ATB has an opportunity to offer that client other financial products with greater profit potential. This is an example of how policy makers need to work with financial institutions and their stakeholders to address the heterogeneous needs of payday borrowers. Together, they need to address the underlying cause of the concern: the lack of unique financial instruments that meet the needs of these borrowers.

While social services, controlled by various levels of government, are often slow and inconvenient as compared to payday lenders, a mindful collaboration between the two groups could provide the crisis group with quick cash in emergencies. For example, when individuals in crisis lose their jobs and have an immediate need to pay a utility bill, they are not able to wait several weeks for social services to provide the needed cash. The details of how this would work are beyond the scope of this research and would require further study.

Even though our discussion has focused on how to address the crisis/unsure group, there are opportunities to service the convenience/unsure group as well. It is unlikely that payday lenders would be interested in directing or encouraging this group to approach mainstream financial institutions for credit. Mainstream financial institutions that actively pursue this group and that provide reasonably priced, convenient-to-obtain products may discover they have attracted a long-term, profitable client.

## 6. Conclusions

We used the TPB in a case study to distinguish attitudes, perceived behavior control, and improvement in the quality of life between borrowers who had difficulty repaying payday loans and those who did not. Consistent with previous research, we found distinct profiles of borrowers that we label crisis and convenience. Whether a borrower had more than one payday loan at the same time was significant in determining the difficulty in repaying the loan, and this characteristic indirectly affected the attitudes and perceived behavior controls. In addition, our research uncovered a previously unidentified group that we refer to as the unsure sub-group. Members of this sub-group, who overlap into both crisis and convenience groups, were unclear about their options for borrowing, their credit rating (perceived behavioral control), and different aspects of payday lenders (attitude). This sub-group of unsure borrowers could potentially be able to access and use a more appropriate form of borrowing if better informed and educated. Given that the payday loan market has distinct borrower profiles, it is clear that meaningful public policy and regulations should reflect the unique needs of each type of borrower, which is unlikely to be achieved with a "one-size-fits-all" legislation.

Our results are similar to Birkenmaier and Fu (2016), Bolen et al. (2020), and Riley et al.'s (2022) observations that an alternative financial service, such as a payday lender, fills

an important niche in the consumer financial marketplace for those who use the service for convenience. However, we also find for the crisis borrower that the regular use of a payday lender for short-term cash needs is costly and potentially harmful for those with few available alternatives—whether real or perceived. For this group, having alternatives to payday loans would be beneficial.

## 7. Limitations and Future Research

Like Mandell and Schmid Klein (2009), and Levinger et al. (2011), we recognize that our sample size is relatively small, and therefore, it is a case study of the specific location and is not representative of the population as a whole. However, we believe that our findings provide a basis for further research into the behaviors and motivations of payday borrowers, particularly those in the unsure category, and they may be useful to adapt to or to provide insight into payday borrowing in other locations. Furthermore, we have limited our analysis to the impacts of attitudes, perceived behavioral controls, and subjective norms among borrowers who have actually had experience with payday loans; therefore, we cannot comment on first-time payday borrowers. We also consider prior behavior and how it could influence the TPB variables. While the main components of the TPB are well studied in other disciplines, previous behavior or the outcome or ultimate effect on the borrower's quality of life have not always been studied. We believe that our adapted framework of the TPB to payday borrowers provides further insights into the descriptive characteristics of each group, and therefore, an opportunity to address the specific needs of all groups: convenience, crisis, and unsure payday borrowers.

Our sample consisted only of individuals with prior payday loan experience; therefore, we measured actual behavior rather than intentions. Future research could investigate the propensity to borrow or the intention to borrow from a payday lender, including both borrowers and non-borrowers, to validate our market profiles.

We cannot eliminate the possibility of reverse causality through more sophisticated statistical methods, as our data were collected during only one period. However, we have used variables representing prior behavior to differentiate between distinct types of borrowers, to attempt to address this concern.

**Author Contributions:** Conceptualization: P.H., I.H., A.K., M.T.; Methodology: I.H., M.T.; Formal Analysis: I.H., F.M.; Writing Original Draft: P.H., I.H., A.K., M.T.; Writing Review and Editing: P.H., I.H., F.M. All authors have read and agreed to the published version of the manuscript.

**Funding:** The CPA Managerial Accounting Fellowship provided financial support (Grant #RT755823); Momentum provided in-kind support.

**Data Availability Statement:** Data are not publicly available due to confidentiality restrictions by the Conjoint Research Ethics Board.

**Conflicts of Interest:** The authors declare no conflict of interests.

## Notes

[1]  Zinman (2010) used household panel data for the state of Oregon in the U.S. Ellison and Forster (2008) used qualitative and quantitative surveys across the Australian market.

[2]  Some advantages of payday loans are the following: no appointment required, minimal requirements to qualify (age of majority, evidence of a job, and bank account); no credit check requirement; convenience (many storefronts and an easy application process); and privacy (the borrowers can avoid disclosure to other financial institutions, joint account holders, family members, etc.).

[3]  Ajzen (1991) found that subjective norms are often weaker than personal considerations, and therefore, they can be poor predictors of behavior, depending on the situation.

[4]  In a saturated model, all the possible paths between the hypothesized variables will be included. While this model is the perfect fit, it might not have proper statistical use. In our study, we compared our model's result with that of the saturated model, and estimated the goodness of fit ($chi^2$), which indicates that our model is a better fit.

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
