# Peer review of "The Paradox of the Payday Borrower: A Case Study of the Role of Planned Behavior in Borrowers’ Motivations and Experiences"

_jrfm, doi:10.3390/jrfm16050254_

Round 1

Reviewer 1 Report

1. In the introductory section, it is important to provide a contextual background for the study.  Presently, it is difficult to know the country or countries which the study is based on. Assuming the study is focused on the U.S, there is a need to provide one or two paragraphs about the state of the payday lending market in the US. This is what I mean by providing a contextual background

2. The structure/organisation of the study should be presented at the end of section 1.

3. In the Methods section, the country context for the study is either not stated or is unclear.. Is the study focused on the US or Canada? The authors should be clear about this.

4. Kindly state the (i) policy implications and (ii) research implications in section 7.

Author Response

Reviewer 1

Comments and Suggestions for Authors

  1. In the introductory section, it is important to provide a contextual background for the study. Presently, it is difficult to know the country or countries which the study is based on. Assuming the study is focused on the U.S, there is a need to provide one or two paragraphs about the state of the payday lending market in the US. This is what I mean by providing a contextual background

Thank you for your suggestions to help us improve the manuscript.

We have included contextual background in both the Introduction and the Methods section and have given additional details about the setting of the study. We hope that this provides additional insight to the study and satisfies the suggestion.

  1. The structure/organisation of the study should be presented at the end of section 1.

We have added a paragraph at the end of Section 1 to indicate the structure of the rest of the paper.

  1. In the Methods section, the country context for the study is either not stated or is unclear.. Is the study focused on the US or Canada? The authors should be clear about this.

The study is focused on a large city in a western Canadian province. We have now indicated this in the manuscript. We have also provided additional details regarding the payday lending context in the province  that motivated the study.

  1. Kindly state the (i) policy implications and (ii) research implications in section 7.

We have elaborated further on policy implications and research implications, although there was considerable discussion of these items in the original version.  You can find policy implications in Discussion and Public Policy Implications and the Conclusions section, as well as other locations in the manuscript.  The Research Implication are also found in the Conclusions but more specifically in Limitations and Future Research. 

Thank you for your suggestions for improving the manuscript.

Reviewer 2 Report

The authors are recommended to make following improvements:

1. Revise the abstract. It is very general in its current form. Please, add details about the research (for instance, the number of respondents)

2. Revise the Introduction. Remove the first paragraphs that a written in the form of an essay. Please, start with the research relevance. Then, define the research gap. The research goal should be clearly stated, as well as methods applied to achieve the goal and test hypotheses. End the Introduction with a brief description of the research results. The description of the TPB should be moved to the Literature review.

3. Consider the possibility to combine the Literature part and the "Conceptual framework". The hypotheses are better formulated in the methodology.

4. Please, substantiate the choice of Factor analysis for 12 items. Is it necessary? What are the results? The factors have not been named. 

5. Please, revise the Bibliography. 31 of 50 sources were published 10 or more years ago. Please, add the recent literature.

Author Response

The authors are recommended to make following improvements:

  1. Revise the abstract. It is very general in its current form. Please, add details about the research (for instance, the number of respondents)

We have included the number of respondents in the abstract and provided more context about the research setting.

  1. Revise the Introduction. Remove the first paragraphs that a written in the form of an essay. Please, start with the research relevance. Then, define the research gap. The research goal should be clearly stated, as well as methods applied to achieve the goal and test hypotheses. End the Introduction with a brief description of the research results. The description of the TPB should be moved to the Literature review.

We have removed the first paragraph from the Introduction. Generally, the theory used is mentioned in the Introduction but developed further in the Literature Review; however, to accommodate the review, we have removed the brief discussion of the Theory of Planned Behavior from the Introduction. We have re-structured the Introduction to start with the research relevance, then we define the research question and the research gap.  We have added more information regarding the methods to test the hypotheses.  The Introduction contains a brief discussion of the findings and ends with how the rest of the manuscript is structured.

  1. Consider the possibility to combine the Literature part and the "Conceptual framework". The hypotheses are better formulated in the methodology.

As suggested by the reviewer, we have combined the Literature and the Conceptual Framework.  The combination does make this section extremely long.

Regarding hypotheses formulated in the Methods section, in our long publishing history, we have never seen the hypotheses in the Methods section.  They should be derived from the Literature and Conceptual Framework. Therefore, we have left the hypotheses development in this section.

  1. Please, substantiate the choice of Factor analysis for 12 items. Is it necessary? What are the results? The factors have not been named. 

Factor Analysis is a pre-requisite to using Structural Equation Modeling. Based on the literature and our previous studies that used these methods, a factor analysis is always completed for data reduction on survey data and to test reliabilities of the extruded factors.  Each factor generally consists of 3-4 questionnaire items. Therefore 12 items consolidated into three factors is consistent with sound statistical procedure. The results of the factor analysis are shown in Table 3. The factors are named in as column headings in Table 3 and interpreted and discussed on p. 22.  The factors are representative of the three main factors in the theory of planned behavior (attitude, behavioral control, and subjective norms).

  1. Please, revise the Bibliography. 31 of 50 sources were published 10 or more years ago. Please, add the recent literature.

We have updated the literature to include more recent citations. Additional citations added are noted below. We hope this satisfies your suggestion.

Agarwal, S. & Bos, M. (2019). Rationality in the consumer credit market: choosing between alternative and mainstream credit. In Haughwout, Andrew and Mandel, Benjamin (eds) Handbook of US Consumer Economics. Elsevier Science and Technology

Bolen, J. B., Elliehausen, G. & Miller, T. (2022). Do consumers need more protection from small-dollar lenders? Historical evidence and a roadmap for future research. Economic Inquiry, 5(4): 1577-1613.

Caplan, M., Kindle, P., and Nielsen, R. (2017). Do we know what we think we know about payday loan borrowers? Evidence from the survey of consumer finances. The Journal of Sociology & Social Welfare, 44(4): Article 3

Riley, L., Green, L., Zuiker, V. (2022). Lived experiences with payday loans: African American single mothers and employees. Family and Consumer Sciences Research Journal, 50(4):301-316.

Thank you for your suggestions to improve the manuscript. I hope we have satisfactorily addressed them.

Round 2

Reviewer 1 Report

Accept.

Author Response

We have added the paragraph shown below to p. 11, which defines our use of financial literacy in our research.  In addition, we have made minor changes to the paragraph that you cite above to explain how the definition is integrated in our research.  We have also made minor changes in other places in the paper.   We hope that this addition adequately addresses your concern.

In the general borrowing literature, attitudes, behavior, knowledge, and confidence have also been identified as important variables in the financial literacy construct (Bialowolski et al. 2020). We build on this literature but focus initially on the application aspect of the definition of financial literacy (capability or access and attitude) to understand why borrowers make the decisions that they do. Then, we use the findings to suggest options for improving their knowledge and skills, and thus their confidence. Rather than testing the general knowledge of the borrowing/lending process, such as interest rates, penalties, and other formal aspects that affect the borrowing decision, we ask the respondent about their more informal perceptions of their specific borrowing decision, such as the extent of their confidence in their lenders’ characteristics and sources of borrowing available to them, and who influences their decisions. We build on both the TPB framework, the payday lending literature, and general borrowing literature to offer insight into unique consumer profiles and suggest that each requires specific consideration to improve well-being and quality of life.